# The Water Storage Function of Litters and Soil in Five Typical Plantations in the Northern and Southern Mountains of Lanzhou, Northwest China

**Shiping Su and Xiaoe Liu ***

College of Forestry, Gansu Agricultural University, Lanzhou 730070, China; susp008@163.com
* Correspondence: liuxiaoe81@126.com

**Abstract:** Soil and water conservation is an important function of forest ecosystems; however, it remains unclear which forest type is best suited for water and soil conservation under the same site conditions. In order to clarify the soil and water conservation function of different plantations in the northern and southern mountains of Lanzhou city, we investigated several soil and water conservation function indicators (thickness and accumulation of litter, maximum water holding capacity and rate of litter, water holding capacity and water absorption rate of litter, soil infiltration rates, soil water content, soil bulk density, soil porosity, and soil water storage) of five plantation types (*Platycladus orientalis* plantations (Po), *Robinia pseudoacacia* plantations (Rp), *Populus alba* var. *pyramidalis* plantations (Pa), *P. alba* var. *pyramidalis* + *R. pseudoacacia* mixed plantations (Pa + Rp), and *P. orientalis* + *R. pseudoacacia* mixed plantations (Po + Rp)) and evaluated them using the gray correlation method. The results indicated the accumulation of litter varied from 13.50 to 47.01 t·hm$^{-2}$ and increased in the order of Pa < Rp < Po < Po + Rp < Pa + Rp. The maximum water holding capacity of litter varied from 35.29 to 123.59 t·hm$^{-2}$ and increased in the order of Pa < Rp < Po < Po + Rp < Pa + Rp. The soil physical properties (soil infiltration, porosity, and bulk density) of mixed plantations were better than those of pure plantations. The soil maximum water storage was significantly different among plantation types ($p < 0.05$), with an average varying from 3930.87 to 4307.45 t·hm$^{-2}$, and was greater in mixed plantations than in pure plantations. Gray correlation analysis revealed that mixed plantations had the best conservation function of the five plantation types, followed by broad-leaved plantations and coniferous plantations. This suggests that the planting of mixed plantations dominated by Pa + Rp is therefore recommended in the future construction of plantations in the northern and southern mountains of Lanzhou to realize sustainable forest development.

**Keywords:** soil and water conservation; plantation types; litter and soil water storage; gray correlation

## 1. Introduction

The forest ecosystems play an important role in regulating regional climate, improving hydrology, conserving water and soil, and conserving water sources in a terrestrial ecosystem. The litter and soil layers store 80–90% of precipitation and are therefore critical in forest water conservation [1]. The litter layer has an important water and soil conservation function in the forest ecosystem; its maximum water holding rate can reach 200.0–448.9% and can intercept 10–20% of precipitation. Litter interception differs depending on stand species' composition, litter type, thickness, storage, water holding capacity, and degree of decomposition of the litter [2–4]. Broad-leaved forest litter usually intercepts more water than coniferous forest litter [5–7]. Forest soil is the main place for water storage in the forest ecosystem. The maximum water storage capacity of forest soil (0–60 cm) in various forest ecosystems is 286.32–486.60 mm, with an average of 383.22 mm, representing 70–80% of precipitation. The soil water storage capacity varies with forest type [8], soil porosity [9,10], soil bulk density [11], soil infiltration rate, and other physical and chemical factors. Therefore, the tree species' composition, decomposition degree of litter, storage and

components of litter, and soil physical and chemical properties jointly determine the water and soil conservation capacity of a forest [12]. In general, the water conservation ability and maximum water holding capacity of the litter layer were stronger in the mixed forests than in the coniferous and broad-leaved forests and were closely related to the accumulation of forest litter and local climate [13]. The soil water storage was greater in the mixed forest than in the pure forest and was closely related to the different soil physical and chemical properties in the different forest types [14–16].

Lanzhou is located in inland China and belongs to the transitional area between the temperate continental and temperate monsoon climate zones [17]. The northern and southern mountains of Lanzhou are important ecological barriers for the city, which is located in the Loess Plateau area. The region has loose soil and steep mountains, making it susceptible to erosion. Since the commencement of afforestation in 1926 [18], 40,000 hm$^2$ of barren mountains and wastelands in the northern and southern mountains have been afforested and a large number of plantations have begun to show ecological benefits. Many researchers have studied the selection of tree species in plantations [19,20], community structure [21], and soil physical and chemical properties [22,23]. However, because of the special climate and site conditions, high-density afforestation was carried out in the early stages of afforestation to promote early canopy closure and early forest formation, which may cause the accumulation and decomposition of litter, water holding capacity, and the improvement of stand on plantation geographical properties, different from other plantations. We hypothesized that the water storage capacity is higher in a mixed plantation than in pure plantations and the mixed plantations have better soil and water conservation function. Therefore, the main objectives of the current study were to investigate the soil and water conservation capacity of plantations in the northern and southern mountains surrounding Lanzhou city; to analyze litter accumulation, decomposition, water holding capacity, soil physical properties, and infiltration characteristics of five typical plantations; and to reveal the water storage function using gray correlation. The results will provide a reference basis for selecting tree species in plantation construction to achieve the optimal effect of soil and water conservation.

## 2. Materials and Methods

### 2.1. Study Sites

The study site was located in the hilly Loess Plateau, which is located in the northern and southern mountains of Lanzhou city, Gansu province (35°53′18″–36°33′56″ N, 103°21′04″–104°00′38″ E). The site was 60 km long from east to west and 5–50 km wide from north to south, with a total area of 42,000 hm$^2$ (Figure 1). The region is characterized by a temperate steppe climate in a semi-arid area, with an annual precipitation of 327.7 mm, an annual evaporation of 1468 mm, and an average annual temperature of 9.1 °C [24]. The soil type is sierozem. The forests in the study area are mainly plantations with natural herbaceous vegetation and a few shrubs. Trees are mainly *Robinia pseudoacacia* L., *Platycladus orientalis* (L.) Franco, *Populus alba* var. *pyramidalis* Bunge, *Pinus tabulaeformis* Carr, and *Sophora japonica* L. [24,25].

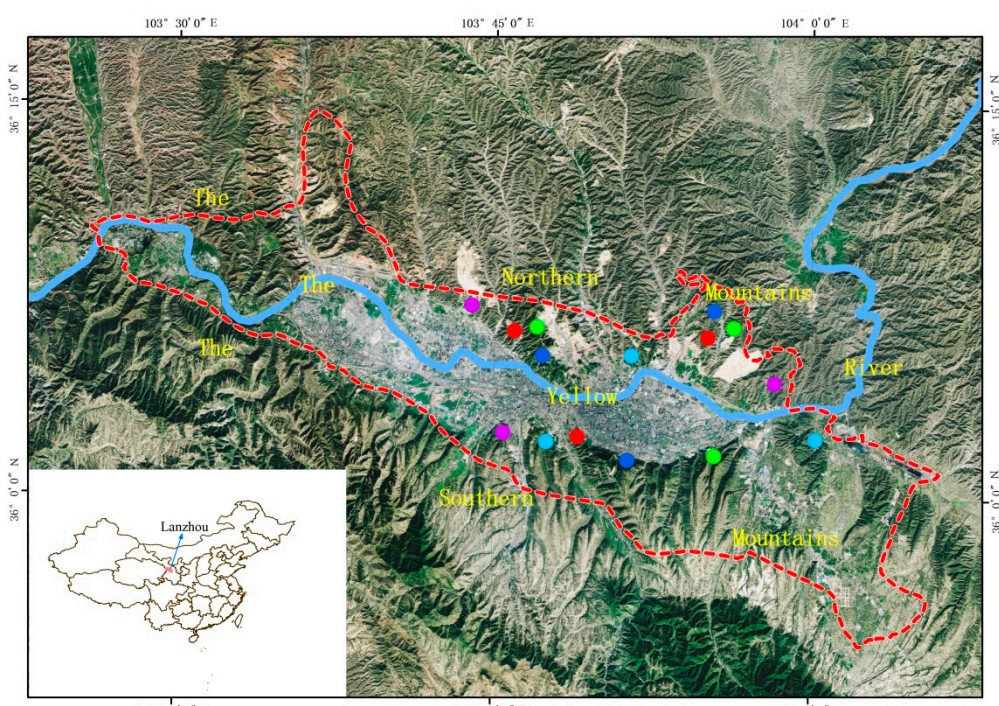

**Figure 1.** Sampling site location of the study stands. ● *P. orientalis* (L.) Franco plantations, ● *R. pseudoacacia* L. plantations, ● *P. alba* var. *pyramidalis* Bunge plantations, ● *P. alba* var. *pyramidalis* Bunge + *R. pseudoacacia* L. mixed plantations, ● *P. orientalis* (L.) Franco + *R. pseudoacacia* L. mixed plantations. Blue line represents the yellow river. The interior of the red line represents the area of the northern and southern mountains.

### *2.2. Experimental Design*

Five plantation types with wide distribution and large area were selected to study the characteristics of litter and soil water holding capacity. (1) *P. orientalis* (L.) Franco plantations (Po): the tree layer is *P. orientalis* (L.) Franco; there are no other trees and shrubs in the plantation. (2) *R. pseudoacacia* L. plantations (Rp): the tree layer is *R. pseudoacacia* L. and a small amount of *Ulmus pumila* L. seedlings; the shrub layer contains a small amount of *Tamarix austrongolica* Nakai. (3) *P. alba* var. *pyramidalis* Bunge plantations (Pa): the tree layer is *P. alba* var. *pyramidalis* Bunge; the shrub layer comprises a small amount of *Amorpha fruticosa* L., *Tamarix chinensis* Lour., *Caragana korshinskii* Kom., and *Elaeagnus angustifolia* L. (4) *P. alba* var. *pyramidalis* Bunge + *R. pseudoacacia* L. mixed plantations (Pa + Rp): the tree layer is composed of *P. alba* var. *pyramidalis* Bunge and *R. pseudoacacia* L., at a species composition ratio of 5:5 (quantity ratio), among which there are a few *U. pumila* L. seedlings; the shrub layer is *T. chinensis* Lour. (5) *P. orientalis* (L.) Franco + *R. pseudoacacia* L. mixed plantations (Po + Rp): the tree layer is composed of *P. orientalis* (L.) Franco and *R. pseudoacacia* L., and the species composition ratio is 3:7 (quantity ratio), among which there are a few *U. pumila* L. seedlings; the shrub layer contains *C. korshinskii* Kom. and *T. austrongolica* Nakai. Three representative standard stands were selected for each stand type, in which three sampling plots with an area of 10 m × 10 m were randomly established to investigate tree species' composition and stand environmental factors (Table 1).

**Table 1.** Basic information of the different stand types.

| Stand Type | Canopy Density | Altitude (m) | Stand Age (Year) | Average Tree Height (m) | Forest Density (Tree·hm$^{-2}$) |
|---|---|---|---|---|---|
| *P. orientalis* | 0.85 | 1813 | 30 | 6.02 | 2977 |
| *R. pseudoacacia* | 0.70 | 1782 | 35 | 9.39 | 2700 |
| *P. alba* var. pyramidalis | 0.60 | 1772 | 35 | 11.06 | 2500 |
| *P. orientalis* + *R. pseudoacacia* | 0.70 | 1761 | 30 | 6.13 | 2740 |
| *P. alba* var. pyramidalis + *R. pseudoacacia* | 0.65 | 1804 | 30 | 10.92 | 2801 |

*2.3. Determination of the Water Holding Capacity of Litter*

To measure the thickness and accumulation of litter in each plot, three small quadrats of 1 m × 1 m along the diagonal direction were randomly selected. The thickness of the litter layer was measured with a steel tape ruler. Subsequently, the living vegetation was removed. Then, the non-decomposed and decomposed litter layers were collected, weighed, and stored in a net bag. The samples were transported to the laboratory and oven-dried at 65 °C to constant weight. Then, the final weight was used to calculate litter accumulation.

In each small quadrat, 50 g of dried litter was put into a 100-mesh nylon net bag (15 cm × 20 cm) and immersed in water. After 0.08, 0.25, 0.5, 1, 1.5, 2, 3, 4.5, 6, 8, 10, 12, 14, 16, 18, 20, 22, and 24 h, the litter was taken out and excessive water was removed. The samples were weighed, then oven-dried at 65 °C to constant weight, then weighed again to determine the water holding capacity [16]. The calculation formula is as follows:

$$R_h = \frac{G_t - G_d}{G_d} \times 100\% \tag{1}$$

$$R_{hmax} = \frac{G_{24} - G_d}{G_d} \times 100\% \tag{2}$$

$$W_h = M \times R_h \tag{3}$$

$$W_{hmax} = M \times R_{hmax} \tag{4}$$

$$V = \frac{G_t - G_d}{t} \tag{5}$$

where $R_h$ is the water holding rate of litter (%), $G_t$ is the wet weight of litter at time $t$ (t·hm$^{-2}$), $G_d$ is the dry weight of litter (t·hm$^{-2}$), $R_{hmax}$ is the maximum water holding rate of litter (%), $G_{24}$ is the litter weight immersion in water for 24 h (t·hm$^{-2}$), $W_{hmax}$ is the maximum water holding capacity of litter (t·hm$^{-2}$), $M$ is the litter accumulation (t·hm$^{-2}$), $W_h$ is the litter water holding capacity (t·hm$^{-2}$), $V$ is the water absorption rate of litter (t·hm$^{-2}$·h$^{-1}$), and $t$ is the immersion time of litter (h).

*2.4. Determination of Soil Physical Properties*

In each plot, three small quadrats with an area of 1 m × 1 m were randomly selected along the diagonal direction to measure soil physical properties. In each small quadrat, a 100 cm$^3$ ring (5.046 cm diameter and 5 cm height) was used to take undisturbed soil at depths of 0–20, 20–40, 40–60, and 60–80 cm, with three replicates per soil layer. The samples were taken back to the laboratory for the measurement of soil infiltration and physical properties.

The soil infiltration rate was measured by the double ring method [26]. In each small quadrat, three undisturbed soil samples were collected with a 100 $cm^3$ ring (5.046 cm diameter and 5 cm height) at 0–20, 20–40, 40–60, and 60–80 cm soil depths. Samples were taken back to the laboratory and immersed in water for 12 h. Then, they were taken out, and an empty ring of same size was added. The interface was sealed with tape and then bonded with melted wax to prevent water leakage. The joint ring was placed on the funnel, with a beaker below the funnel. Water was added to the ring on the upper side with a thickness of 5 cm. After the water addition, timing began when the first drop of water fell from the bottom of the funnel. Then, the beaker under the funnel was replaced after 1 min, 2 min, 3 min, 4 min, 5 min, and every 5 min until 60 min to measure the exudation water ($Q_1$, $Q_3$, $Q_6$, $Q_{10}$, $Q_{15}$, $Q_{20}$ ... $Q_n$). After each change of the beaker, the water surface of the upper ring to the original height until the leakage water was equal. In this study, the leakage water was considered stable when the experiment had been running for 60 min.

Soil bulk density, total porosity, and capillary porosity were measured by the ring method [27]. The soil water content was determined by the drying method [27]:

$$W_m = 100P_m h \tag{6}$$

$$W_o = 100P_o h \tag{7}$$

$$W_c = 100P_c h \tag{8}$$

where $W_m$ is the maximum soil water holding capacity (t·$hm^{-2}$), $W_o$ is the soil non-capillary water holding capacity (t·$hm^{-2}$), $W_c$ is the soil capillary water holding capacity (t·$hm^{-2}$), $P_m$ is the total soil porosity (%), $P_o$ is the soil non-capillary porosity (%), $P_c$ is the soil capillary porosity (%), and $h$ is the soil thickness (cm).

### 2.5. Using the Gray Correlation Method to Evaluate the Water and Soil Conservation Capacity

The gray correlation method is a relatively simple and reliable approach because it has no requirement on sample number and low computation and does not follow typical distribution rules [28–30]. It is, therefore, widely used in studies on forest soil and water conservation capacity [16].

Firstly, the reference sequence $Y_0$ and comparison sequence $Y_i$ were established. In the five stand types, the reference sequence $Y_0$ was established based on the value slightly higher than the maximum value of each measured index, and the comparison sequence $Y_i$ was established based on the measured value of each index in each stand.

$$Y_0 = \{Y_0(1), Y_0(2), Y_0(3), \dots , Y_0(n)\}$$

$$Y_i = \{Y_i(1), Y_i(2), Y_i(3), \dots , Y_i(n)\}, i = 1, 2, 3, \dots \dots , m$$

Then, in order to eliminate the dimensionality of the data, the $Y_0$ and $Y_i$ sequences were standardized and the standardized data sequences $Y_0(k)$ and $Y_i(k)$ were established:

$$Y_0(k) = \frac{Y_0(n)}{Y_0(n)}, Y_i(k) = \frac{Y_0(n)}{Y_i(n)} \tag{9}$$

The absolute value of the difference $\Delta_i(k)$ between the reference sequence $Y_0(k)$ and the comparison sequence $Y_i(k)$ at each corresponding point were calculated to establish the difference sequence $\Delta_i$:

$$\Delta_i(k) = |Y_0(k) - Y_i(k)| \tag{10}$$

Finally, the *maximum*$\Delta_i(k)_{max}$ and *minimum*$\Delta_i(k)_{min}$ of the $\Delta_i$ sequence were used to determine the correlation coefficient $\xi_i(k)$ and establish the $\xi_i$ data sequence:

$$\xi_i(k) = \frac{minmin\Delta_i(k) + \rho maxmax\Delta_i(k)}{\Delta_i(k) + \rho maxmax\Delta_i(k)} \tag{11}$$

where $\rho$ is the resolution coefficient ranging from 0–1 and $minmin\Delta_i(k)$ and $maxmax\Delta_i(k)$ are the maximum and minimum values of the $\Delta_i(k)$ in each point, respectively. Here, $\rho$ was artificially set to 0.5. The formula of $r_i$ can be expressed as

$$r_i = \frac{1}{m} \sum_{k=1}^{m} \xi_i(k) \tag{12}$$

where $\xi_i(k)$ and $r_i$ are the gray correlation coefficient and the degree of the gray correlation, respectively.

The coefficient of variation $W_i$ was calculated as follows:

$$W_i = \frac{r_i}{\sum_{k=1}^{n} r_i} \tag{13}$$

Subsequently, we calculated $G_k$ with the correlation degree value:

$$G_k = \sum_{k=1}^{n} \xi_i(k) W_i \tag{14}$$

where $G_k$ is the gray comprehensive evaluation value.

### 2.6. Statistical Analysis

The statistical software package SPSS 17.0 was used for one-way AVOVA data analysis and multiple comparisons (Duncan at 0.05 level). Microsoft Excel 2010 software was used for gray correlation analysis.

## 3. Results
### 3.1. Water Holding Characteristics of Litter Layer under Different Stand Types
3.1.1. Thickness and Accumulation of Litter

The thickness and accumulation of litter were significantly different among the stand types ($p < 0.05$). The litter thickness in the five stand types varied from 2.08 to 5.52 cm, with the highest level for Pa + Rp and the lowest level for Po; there was a significant difference between the two stands ($p < 0.05$) (Table 2). The accumulation of litter varied from 13.50 to 47.01 t·hm$^{-2}$ and increased in the order of Pa < Rp < Po < Po + Rp < Pa + Rp. The accumulation of litter in the Pa + Rp was 3.48 times that of the Pa; there were significant differences among all other stand types ($p < 0.05$), except between Po and Rp ($p > 0.05$) (Table 2).

In the five stand types, the proportion of the undecomposed and semi-decomposed layer accumulations in the total accumulation of litter varied among the stand types; the proportion of the semi-decomposed layer accumulation varied from 56.3% to 66.5%. Except for Po, the proportion of the semi-decomposed layer accumulation was significantly higher than that of the undecomposed layer. Among the undecomposed litter samples of the five stand types, the proportion of the undecomposed layer accumulation was lowest in the Pa + Rp (33.5%) and was highest in the Po (54.5%) (Table 2).

**Table 2.** Litter characteristics of different stand types.

| Stand Types | Litter Thickness (mm) | Litter Accumulation (t·hm$^{-2}$) | | | Maximum Water Holding Capacity (t·hm$^{-2}$) | | | Maximum Water Holding Rate (%) | | |
|---|---|---|---|---|---|---|---|---|---|---|
| | | Undecomposed Layer | Semi-Decomposed Layer | Total | Undecomposed Layer | Semi-Decomposed Layer | Total | Undecomposed Layer | Semi-Decomposed Layer | Average |
| *P. orientalis* | 2.08 ± 0.37 c | 15.70 ± 1.70 a | 13.08 ± 0.96 d | 28.78 ± 2.58 c | 25.35 ± 3.77 c | 29.62 ± 2.46 c | 54.98 ± 6.03 c | 161.1 ± 7.0 d | 226.4 ± 3.5 a | 190.8 ± 4.6 b |
| *R. pseudoacacia* | 3.81 ± 0.37 b | 9.12 ± 1.25 b | 17.22 ± 1.86 c | 26.34 ± 3.03 c | 24.74 ± 0.75 c | 38.19 ± 4.25 bc | 62.93 ± 4.86 c | 273.7 ± 27.4 b | 222.8 ± 25.7 a | 240.0 ± 19.6 a |
| *P. alba* var. *pyramidalis* | 4.43 ± 0.12 b | 5.41 ± 0.37 c | 8.09 ± 0.93 e | 13.50 ± 0.95 d | 16.59 ± 1.25 d | 18.71 ± 2.62 d | 35.29 ± 3.43 d | 308.4 ± 42.9 ab | 231.5 ± 19.9 a | 262.1 ± 28.5 a |
| *P. orientalis* + *R. pseudoacacia* | 4.07 ± 0.49 b | 15.37 ± 0.55 a | 19.78 ± 0.32 b | 35.15 ± 0.23 b | 33.82 ± 2.77 b | 46.31 ± 6.24 b | 80.13 ± 8.66 b | 220.7 ± 25.3 c | 233.9 ± 27.8 a | 228.1 ± 26.1 ab |
| *P. alba* var. *pyramidalis* + *R. pseudoacacia* | 5.52 ± 0.90 a | 15.75 ± 0.35 a | 31.26 ± 1.03 a | 47.01 ± 1.16 a | 54.95 ± 2.09 a | 68.64 ± 9.88 a | 123.59 ± 11.73 a | 349.0 ± 18.4 a | 219.1 ± 24.9 a | 262.7 ± 20.0 a |

Lowercase letters indicate that the same index is significantly different among different stand types ($p < 0.05$).

### 3.1.2. Maximum Water Holding Capacity and Maximum Water Holding Rate of Litter

The water storage function of a stand is closely related to the water holding capacity of the stand litter. The water holding capacity of litter varied significantly among the stand types ($p < 0.05$) (Table 2). The maximum water holding rate of litter from the five stand types varied from 190.8% to 262.7% and increased in the order of Po < Po + Rp < Rp < Pa < Pa + Rp; the maximum water holding rate of Pa + Rp was 1.38 times that of Po (Table 2). The maximum water holding capacity of litter from the five stand types varied from 35.29 to 123.59 t·hm$^{-2}$ and increased in the order of Pa < Rp < Po < Po + Rp < Pa + Rp; the maximum water holding capacity of Pa + Rp was 3.50 times that of Pa (Table 2).

The water holding capacity of different litter layers varied with the stand type (Table 2). The maximum water holding rate of the undecomposed layer was significantly different among the stand types ($p < 0.05$) and increased in the order of Po < Po + Rp < Rp < Pa < Pa + Rp. The maximum water holding capacity of the undecomposed layer was significantly different among the stand types ($p < 0.05$) and increased in the order of Pa < Rp < Po < Po + Rp < Pa + Rp. There was no significant difference in the maximum water holding rate of the semi-decomposed layer among the stand types ($p > 0.05$) but there was a significant difference in the maximum water holding capacity among the stand types ($p < 0.05$). The maximum water holding capacity increased in the order of Pa < Po < Rp < Po + Rp < Pa + Rp (Table 2).

### 3.1.3. Water Holding Capacity and Water Absorption Rate of Litter

The water holding capacity of litter from the stand types increased initially and then stabilized with the extension of the immersion time. Within 2 h of the initial immersion, the values increased rapidly; but the capacity decreased after 2–8 h and reached saturation after 12 h (Figure 2). The water holding capacity of litter in the undecomposed layer was lower than that in the semi-decomposed layer. The water holding capacity of the undecomposed layer increased in the order of Pa < Rp < Po < Po + Rp < Pa + Rp; however, the water holding capacity of the semi-decomposed layer increased in the order of Pa < Po < Rp < Po + Rp < Pa + Rp (Figure 2).

The variation trend of the litter water absorption rate with the immersing time remained consistent among the stand types (Figure 2). Within 1 to 2 h of the initial immersion, the water absorption rate decreased linearly then gradually slowed down, reaching a consistent rate 10 h later when the water absorption of litter was close to saturation. The water absorption rate of litter in the undecomposed layer was lower than that in the semi-decomposed layer. Within 5 min of the initial immersion, the water absorption rate of the undecomposed layer varied from 95.89 to 229.95 t·hm$^{-2}$·h$^{-1}$ and increased in the order of Po < Pa < Rp < Po + Rp < Pa + Rp. The water absorption rate of the semi-decomposed layer varied from 111.78 to 477.65 t·hm$^{-2}$·h$^{-1}$ and increased in the order of Pa < Po < Rp < Po + Rp < Pa + Rp (Figure 2).

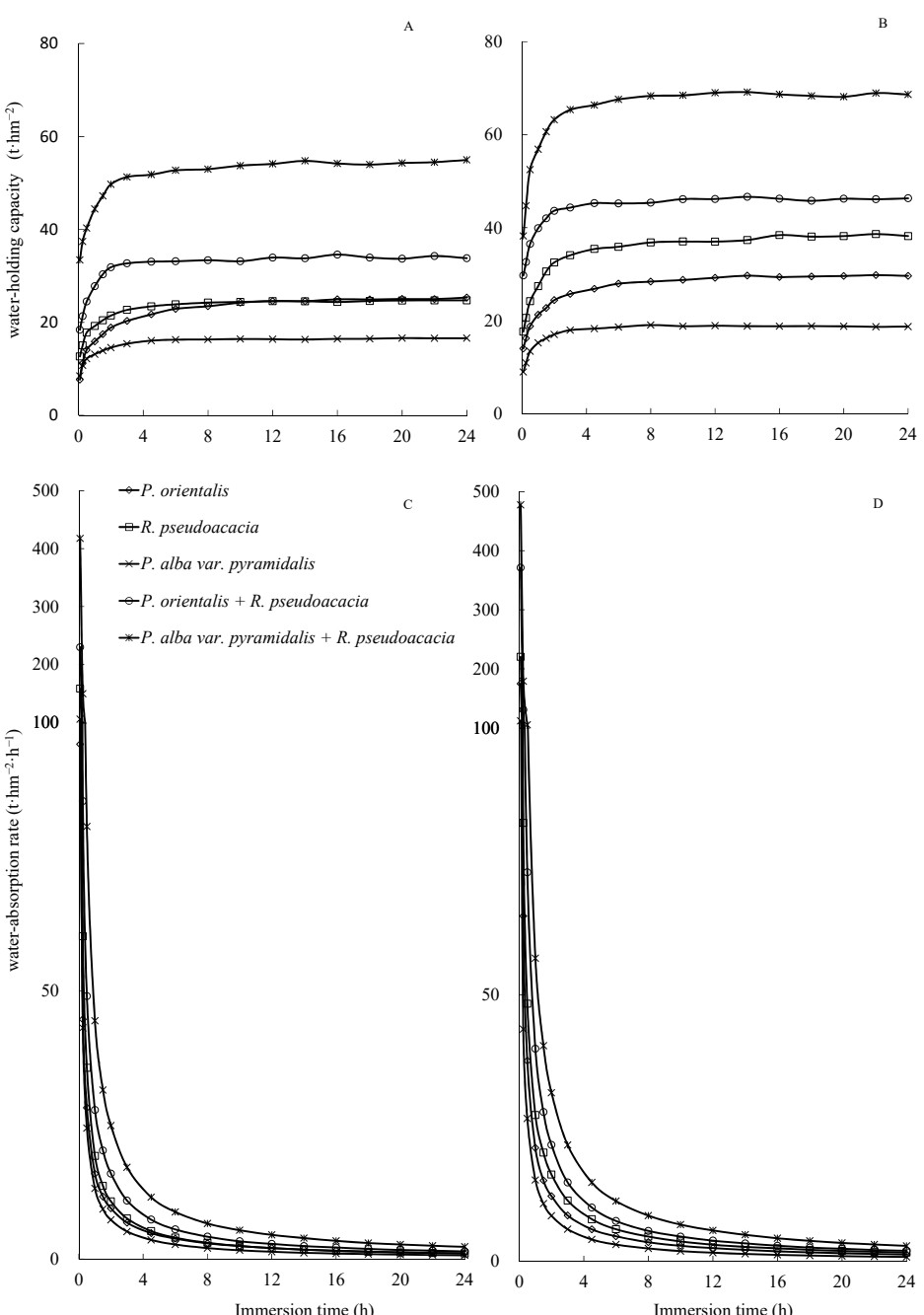

**Figure 2.** Water holding capacity and water absorption rate of litter under different stand types; (**A**,**C**) represent the undecomposed layer, (**B**,**D**) represent semi-decomposed layer.

### 3.2. Soil Physical Properties under Different Stand Types

#### 3.2.1. Soil Infiltration Rates

The soil infiltration rate under all the stand types showed a downward trend with the increase in soil depth from 0 to 80 cm (Figure 3). The soil infiltration rate varied among the stand types; the differences gradually decreased with the increase in the soil depth. The soil infiltration rates increased in the order of Pa < Po < Rp < Po + Rp < Pa + Rp at 0–20 and 20–40 cm soil depths. However, the soil infiltration rate of Pa + Rp was higher than that of the other four stand types at 40–60 cm soil depth; there was no significant difference among the other four stand types. There was no significant difference among the stand types at 60–80 cm soil depth (Figure 3).

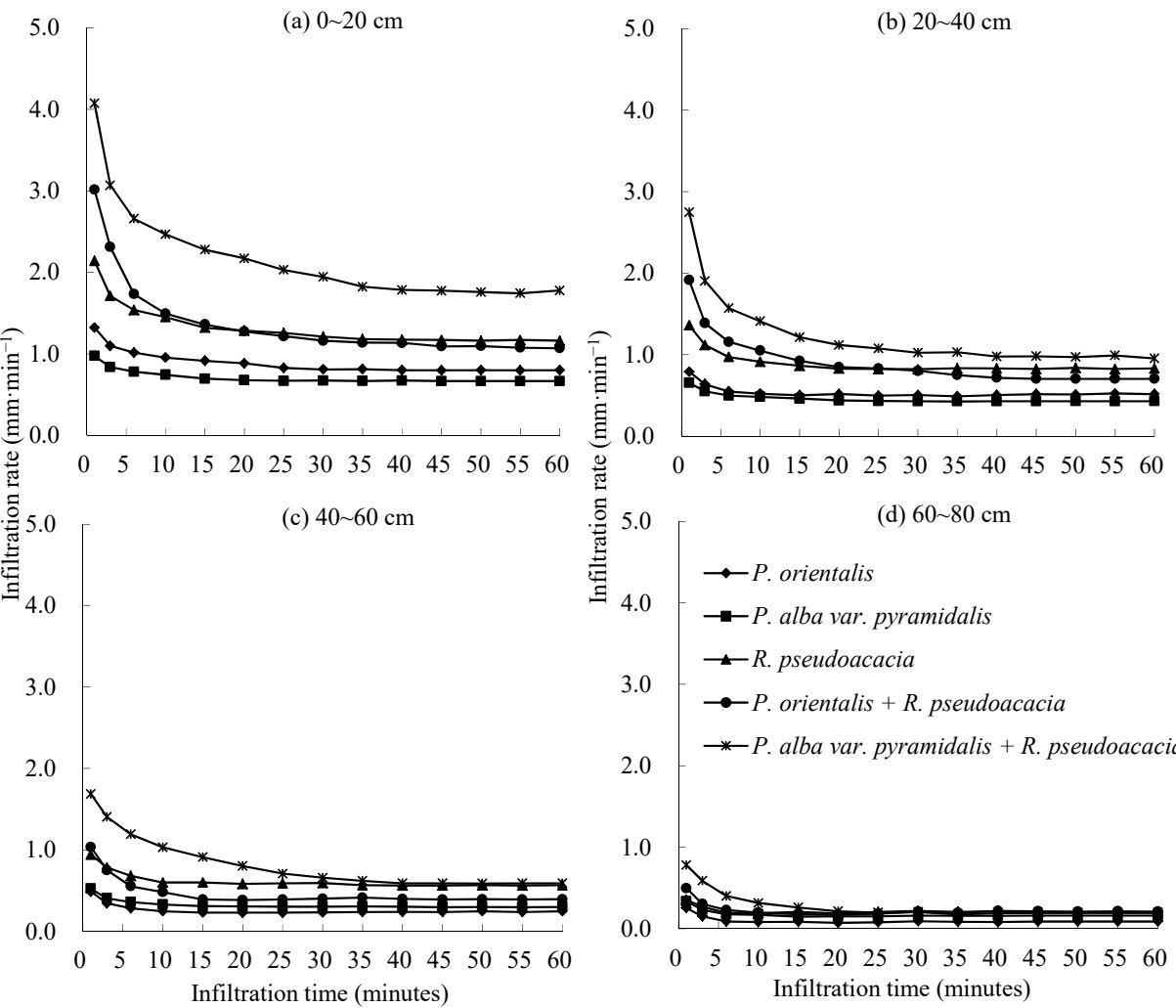

**Figure 3.** Soil infiltration rate of the different stand types.

3.2.2. Soil Water Content

The soil water content differed significantly among the five stand types ($p < 0.05$) (Table 3). The soil water content of the five stand types varied from 4.36% to 13.82% and was higher in mixed stand types than in pure stand types. The soil water content was highest in Pa + Rp (13.82%), followed by Po + Rp (12.73%). The soil water content of Po, Pa, and Rp were 5.96%, 7.96%, and 4.36%, respectively, which were 56.87%, 42.40%, and 68.45% lower than that of Pa + Rp, respectively (Table 3).

**Table 3.** Physical properties in the different stand types.

| Stand Types | Soil Layer (cm) | Soil Bulk Density (g·cm$^{-3}$) | Soil Water Content (%) | Capillary Porosity (%) | Non-Capillary Porosity (%) | Total Porosity (%) | Maximum Water Storage (t·hm$^{-2}$) | Non-Capillary Water Storage (t·hm$^{-2}$) | Capillary Water Storage (t·hm$^{-2}$) |
|---|---|---|---|---|---|---|---|---|---|
| *P. orientalis* | 0–20 | 1.24 ± 0.03 c | 7.38 ± 0.41 a | 46.67 ± 0.90 a | 6.49 ± 0.24 a | 53.16 ± 1.07 a | 1063.22 ± 21.4 a | 129.82 ± 4.8 a | 933.40 ± 18 a |
| | 20–40 | 1.30 ± 0.06 bc | 6.27 ± 0.56 a | 45.92 ± 1.18ab | 5.29 ± 0.68 a | 51.21 ± 1.86 ab | 1024.22 ± 37.2 ab | 105.84 ± 13.6 a | 918.38 ± 23.6 ab |
| | 40–60 | 1.42 ± 0.05 ab | 6.17 ± 0.54 a | 41.79 ± 2.09 bc | 5.17 ± 0.68 a | 46.97 ± 1.67 bc | 939.28 ± 33.4 bc | 103.43 ± 13.6 a | 835.85 ± 41.8 bc |
| | 60–80 | 1.48 ± 0.04 a | 4.01 ± 0.97 b | 40.34 ± 0.90 c | 4.87 ± 0.62 a | 45.21 ± 1.37 c | 904.14 ± 27.4 c | 97.34 ± 12.4 a | 806.80 ± 18 c |
| | Mean/sum | 1.36 ± 0.03 | 5.96 ± 0.08 | 43.68 ± 1.03 | 5.46 ± 0.35 | 49.14 ± 1.11 | 3930.87 | 436.44 | 3494.43 |
| *P. alba* var. *pyramidalis* | 0–20 | 1.21 ± 0.07 b | 5.23 ± 0.50 a | 48.02 ± 2.87 a | 6.07 ± 0.79 a | 54.09 ± 2.42 a | 1081.77 ± 48.4 a | 121.40 ± 15.8 a | 960.37 ± 57.4 a |
| | 20–40 | 1.36 ± 0.05 ab | 4.74 ± 0.18 ab | 43.61 ± 1.09 ab | 5.52 ± 1.16 a | 49.14 ± 1.66 ab | 982.75 ± 33.2 ab | 110.44 ± 23.2 a | 872.31 ± 21.8 ab |
| | 40–60 | 1.41 ± 0.01 a | 3.87 ± 0.48 ab | 42.30 ± 0.78 ab | 4.98 ± 0.24 a | 47.29 ± 0.54 b | 945.81 ± 10.8 b | 99.71 ± 4.8 a | 846.10 ± 15.6 ab |
| | 60–80 | 1.44 ± 0.03 a | 3.59 ± 0.39 b | 41.91 ± 1.44 b | 4.73 ± 0.51 a | 46.63 ± 0.98 b | 986.20 ± 19.6 b | 135.18 ± 10.2 a | 851.02 ± 28.8 b |
| | Mean/sum | 1.33 ± 0.05 | 4.36 ± 0.11 | 44.12 ± 0.74 | 5.33 ± 0.38 | 49.96 ± 0.98 | 3996.54 | 466.74 | 3529.80 |
| *R. pseudoacacia* | 0–20 | 1.25 ± 0.05 a | 11.92 ± 0.58 a | 44.95 ± 1.07 a | 7.83 ± 0.56 a | 52.78 ± 1.62 a | 1055.58 ± 32.4 a | 156.54 ± 11.2 a | 899.04 ± 21.4 a |
| | 20–40 | 1.28 ± 0.06 a | 7.23 ± 1.26 b | 44.25 ± 1.49 a | 7.53 ± 0.73 a | 51.78 ± 1.89 a | 1035.54 ± 37.8 a | 150.59 ± 14.6 a | 884.95 ± 29.8 a |
| | 40–60 | 1.30 ± 0.09 a | 5.59 ± 0.81 b | 44.26 ± 1.41 a | 6.72 ± 1.98 a | 50.99 ± 2.88 a | 1019.79 ± 57.6 a | 134.54 ± 39.6 a | 885.25 ± 28.2 a |
| | 60–80 | 1.35 ± 0.04 a | 7.09 ± 1.38 b | 42.55 ± 0.38 a | 6.76 ± 1.62 a | 49.31 ± 1.24 a | 986.20 ± 24.8 a | 135.18 ± 32.4 a | 851.02 ± 7.6 a |
| | Mean/sum | 1.30 ± 0.03 | 7.96 ± 0.50 | 44.00 ± 0.78 | 7.21 ± 1.19 | 51.21 ± 1.80 | 4097.11 | 576.85 | 3520.26 |
| *P. orientalis* + *R. pseudoacacia* | 0–20 | 1.11 ± 0.04 b | 15.20 ± 0.71 a | 49.08 ± 0.80 a | 8.22 ± 0.38 a | 57.29 ± 1.17 a | 1145.80 ± 23.4 a | 164.30 ± 34 a | 981.49 ± 22 a |
| | 20–40 | 1.25 ± 0.02 a | 13.66 ± 0.07 ab | 45.33 ± 0.91 ab | 7.28 ± 1.49 a | 52.61 ± 0.59 b | 1052.21 ± 11.8 b | 145.55 ± 11.2 a | 906.67 ± 3.2 ab |
| | 40–60 | 1.31 ± 0.04 a | 12.36 ± 0.56 b | 44.70 ± 1.44 b | 6.23 ± 0.95 a | 50.93 ± 1.32 b | 1018.64 ± 26.4 b | 124.65 ± 16.2 a | 893.99 ± 15.6 b |
| | 60–80 | 1.34 ± 0.03 a | 9.69 ± 0.32 c | 43.69 ± 1.48 b | 5.99 ± 0.48 a | 49.68 ± 1.04 b | 993.52 ± 20.8 b | 119.75 ± 16.4 a | 873.77 ± 13.4 b |
| | Mean/sum | 1.25 ± 0.01 | 12.73 ± 0.18 | 45.70 ± 0.85 | 6.93 ± 0.70 | 52.63 ± 0.14 | 4210.18 | 554.26 | 3655.92 |
| *P. alba* var. *pyramidalis* + *R. pseudoacacia* | 0–20 | 1.05 ± 0.02 d | 17.00 ± 0.28 a | 49.63 ± 1.10 a | 9.60 ± 1.70 a | 59.23 ± 0.69 a | 1184.51 ± 13.8 a | 191.94 ± 7.6 a | 992.57 ± 16 a |
| | 20–40 | 1.16 ± 0.02 c | 15.13 ± 0.58 b | 47.92 ± 0.16 ab | 7.70 ± 0.56 ab | 55.62 ± 0.64 b | 1112.34 ± 12.8 b | 153.96 ± 29.8 ab | 958.39 ± 18.2 ab |
| | 40–60 | 1.26 ± 0.02 b | 13.62 ± 0.15 c | 45.48 ± 0.78 b | 6.80 ± 0.81 ab | 52.28 ± 0.52 c | 1045.60 ± 10.4 c | 135.96 ± 19.0 ab | 909.65 ± 28.8 b |
| | 60–80 | 1.39 ± 0.02 a | 9.52 ± 0.62 d | 42.44 ± 0.67 c | 5.81 ± 0.82 b | 48.25 ± 0.64 d | 964.99 ± 12.8 d | 116.20 ± 9.6 b | 848.79 ± 29.6 c |
| | Mean/sum | 1.22 ± 0.01 | 13.82 ± 0.26 | 46.37 ± 0.05 | 7.48 ± 0.51 | 53.84 ± 0.49 | 4307.45 | 598.06 | 3709.39 |

Different lowercase letters in the same column indicate significant differences among different soil depths of the same stand type ($p < 0.05$).

### 3.2.3. Soil Bulk Density

There were significant differences in soil bulk density among the stand types ($p < 0.05$) (Table 3), with an average varying from 1.22 to 1.36 g·cm$^{-3}$. The soil bulk density increased in the order of Pa + Rp < Po + Rp < Pa < Rp < Po. The soil bulk density was largest in Po (1.36 g·cm$^{-3}$), which was 1.11 times that of Pa + Rp, and the soil bulk density was lower in the mixed stand types than in the pure stand types. The soil bulk density increased with the soil depth, with an average varying from 1.17 to 1.40 g·cm$^{-3}$, and the difference between the stand types was significant at 0–80 cm soil depth ($p < 0.05$) (Table 3).The soil bulk density of 60–80 cm soil depth was 1.04, 1.10, and 1.20 times that at 40–60, 20–40, and 0–20 cm soil depths, respectively (Table 3).

### 3.2.4. Soil Porosity

The total soil porosity was significantly different among the stand types ($p < 0.05$) (Table 3), with an average varying from 49.14% to 53.84%, and increasing in the order of Po < Rp < Pa < Po + Rp < Pa + Rp. The total soil porosity was the highest in Pa + Rp, accounting for 53.84%, which was 1.10 times that of Po, and the total soil porosity was higher in mixed stand types than in pure stand types. The total soil porosity decreased with the soil depth, with an average between 47.82% and 55.31%; the difference was significant among different soil depths ($p < 0.05$) (Table 3). The total porosity at 0–20 cm soil depth was 1.06, 1.11, and 1.16 times that at 20–40, 40–60, and 60–80 cm soil depths, respectively (Table 3).

The soil capillary porosity was significantly different among stand types ($p < 0.05$) (Table 3), with an average varying from 43.68% to 46.37%, and it increased in the order of Po < Rp < Pa < Po + Rp < Pa + Rp. The soil capillary porosity was highest in Pa + Rp, which was 1.06 times that of Po, and the soil capillary porosity was higher in the mixed stand types than in the pure stand types. The soil capillary porosity decreased with the increase in soil depth, with an average varying from 42.19% to 47.67%; the difference in 0–80 cm soil depth was significant ($p < 0.05$) (Table 3). The capillary porosity at 0–20 cm soil depth was 1.05, 1.09, and 1.13 times that at 20–40, 40–60, and 60–80 cm soil depths, respectively.

The soil non-capillary porosity had a significant difference among the stand types ($p < 0.05$) (Table 3), with an average varying from 5.33% to 7.48%. The soil non-capillary porosity decreased with soil depth, with an average varying from 5.63% to 7.64%; the difference in 0–80 cm soil depth was significant ($p < 0.05$) (Table 3). The non-capillary porosity at 0–20 cm soil depth was 1.15, 1.28, and 1.36 times that at 20–40, 40–60, and 60–80 cm soil depths, respectively.

### 3.2.5. Soil Water Storage

The soil maximum water storage was significantly variable among stand types ($p < 0.05$) (Table 3), with an average between 3930.87 and 4307.45 t·hm$^{-2}$. The maximum soil water storage was highest in Pa + Rp and lowest in Po. The maximum water storage was greater in the mixed stand types than in the pure stand types. The soil maximum water storage decreased with the soil depth; the difference at 0–80 cm soil depth was significant ($p < 0.05$) (Table 3).

The soil capillary water storage varied significantly among the stand types ($p < 0.05$) (Table 3), with an average varying from 3494.43 to 3709.39 t·hm$^{-2}$. The soil capillary water storage was highest in Pa + Rp and lowest in Po; the soil capillary water storage was greater in the mixed stand types than in the pure stand types and greater in the broad-leaved stand types than in the coniferous stand types. The soil maximum water storage decreased with the soil depth; the difference at 0–80 cm soil depth was significant ($p < 0.05$) (Table 3).

There was no significant difference in non-capillary water storage among the stand types ($p > 0.05$), with an average varying from 436.44 to 598.06 t·hm$^{-2}$; the difference at 0–80 cm soil depth was significant ($p < 0.05$) (Table 3). The soil non-capillary water storage increased in the order of Po < Pa < Po + Rp < Rp < Pa + Rp.

### 3.3. Soil and Water Conservation Function under Different Stand Types

The comprehensive evaluation value directly reflects the water conservation capacity of different stand types [31]. The larger the comprehensive evaluation value is, the higher the effect of the corresponding water conservation capacity is [32]. According to Table 4, the conservation function of the five forest sites increased in the order of Po < Pa < Rp < Po + Rp < Pa + Rp. This indicates that the water and soil conservation function of the mixed stand types was the best, followed by that of the broad-leaved stand types and coniferous stand types. The correlation degree decreased in the order of soil capillary water storage > soil bulk density > soil non−capillary water storage > maximum water holding rate of litter > soil water content > litter accumulation > maximum water holding capacity of litter> soil infiltration rate. The soil capillary water storage, non-capillary water storage, and bulk density were identified as the most important factors affecting soil water and soil conservation capacity, while the soil infiltration rate had the least influence.

**Table 4.** Correlation degrees and correlation coefficients of each evaluation index.

| Stand Types | Litter Water Holding Capacity | Litter Water Holding Rate | Soil Infiltration Rate | Soil Water Content | Capillary Water Storage | Non-Capillary Water Storage | Soil Bulk Density | Litter Accumulation | $G_k$ |
|---|---|---|---|---|---|---|---|---|---|
| *P. orientalis* | 0.3796 | 0.5166 | 0.3545 | 0.3915 | 0.8332 | 0.5887 | 0.9353 | 0.4780 | 0.6072 |
| *R. pseudoacacia* | 0.4005 | 0.6621 | 0.4395 | 0.4527 | 0.8456 | 0.9160 | 0.8433 | 0.4507 | 0.6760 |
| *P. alba* var. *pyramidalis* | 0.3362 | 0.7580 | 0.3454 | 0.3532 | 0.8503 | 0.6379 | 0.9279 | 0.3466 | 0.6250 |
| *P. orientalis* + *R. pseudoacacia* | 0.4546 | 0.6199 | 0.4319 | 0.7225 | 0.9175 | 0.8408 | 0.7904 | 0.5676 | 0.7076 |
| *P. alba* var. *pyramidalis* Bunge + *R. pseudoacacia* | 0.6905 | 0.7610 | 0.6168 | 0.8364 | 0.9493 | 1.0000 | 0.7499 | 0.8721 | 0.8293 |
| $r_i$ | 0.4523 | 0.6635 | 0.4376 | 0.5513 | 0.8792 | 0.7967 | 0.8493 | 0.5430 | |
| $W_i$ | 0.0874 | 0.1283 | 0.0846 | 0.1066 | 0.1700 | 0.1540 | 0.1642 | 0.1050 | |

## 4. Discussion

The accumulation of forest litter reflects the productivity of the forest community and the litter decomposition degree [33]. The more litter accumulates, the stronger the water and soil conservation function of the stand. We found that the litter accumulation was higher in the mixed stand types than in the pure stand types in the five stand types and was largest in the broad-leaved mixed stand types. He et al. [34] also demonstrated that the litter accumulation in mixed stands was higher than that in pure stands. Chen et al. stated that the litter accumulation was higher in broad-leaved stands than in coniferous stands [35]. This may be because of the non-consistent growth rate of various tree species in mixed stand types, resulting in different canopy levels, thus increasing the light receiving area of the canopy and improving the utilization rate of light energy. This is similar to a multi-layer plantation. However, the canopy of a pure stand type remains at the same level; thus, the plants block light from each other. In addition, *R. pseudoacacia* is a leguminous nitrogen-fixing tree species. Nitrogen-fixing bacteria can form a symbiotic nitrogen-fixing system with *R. pseudoacacia* roots, thus increasing soil fertility and providing more nutrients for the growth of *P. alba* var. *pyramidalis* [36].

The water holding rate of litter is related to forest type, litter characteristics, and decomposition degree [37]. This study showed that the litter water holding rate of Po and Po + Rp in the undecomposed layer were lower than those in the semi-decomposed layer. This was because the undecomposed litter of *P. orientalis* contains oil and, thus, has a poor hydrophilicity. However, after decomposition, the oil content decreases and the water holding rate will increase accordingly [2].

Litter water holding capacity is determined by the water holding rate and accumulation. We found that the litter water holding capacity of the undecomposed layer was less

than that of the semi-decomposed layer because the semi-decomposed layer had higher litter accumulation. The water holding capacity of the different stand types was the highest in Pa + Rp, followed by Po + Rp, and was lowest in Pa. Similar results were observed previously [34]. Zheng et al. also demonstrated that the litter water holding capacity was higher in mixed stand types than in pure stand types [16]. This was caused by the higher accumulation and water holding capacity in mixed stand types. Although Pa has a larger litter water holding rate, its litter water holding capacity is smaller because of its lower litter accumulation [38].

The water absorption rate of litter reflects its ability to intercept precipitation within a short time, which is positively correlated with the water holding capacity. At a high water absorption rate, the litter layer absorbs precipitation quickly, thus effectively reducing the formation of surface runoff [39]. In the current study, we found that the water absorption rate of litter in the undecomposed layer was less than that in the semi-decomposed layer because the water absorption rate increased after the litter of Po and Po + Rp was decomposed and the oil content of *P. orientalis* litter was reduced. However, the water absorption rate was correspondingly higher in the semi-decomposed layer than in the other three stand types as a result of the higher litter accumulation [2].

Different stand types have different soil physical properties. In this study, the soil infiltration, bulk density, and porosity of the mixed stand types were better than those of the pure stand types. Wei et al. also found that soil physical properties were better in the mixed stand types than in the pure stand types [40]. This was because the roots of the different tree species are distributed at different soil depths in mixed stand types. Studies have shown that the vast majority of *R. pseudoacacia* roots are concentrated in the 0–120 cm soil layer [41,42], whereas more than 80% of *P. alba* var. *pyramidalis* roots are distributed in the 0–40 cm soil layer [43] and *P. orientalis* roots are mainly distributed in the 0–50 cm soil layer [44]. Therefore, mixed stand types increase the absorption space of soil nutrients; increased root extension can increase soil permeability and porosity and reduce soil bulk density. Differences in litter accumulation also lead to variation in soil physical properties among stand types. We found that the litter accumulation was 35.15–47.01 t·hm$^{-2}$ in the mixed stand types, whereas it was 13.5–28.78 t·hm$^{-2}$ in the pure stand types. We found that increased litter decomposes to form increased soil organic matter (10.14–11.67 g·kg$^{-1}$) [45], and the soil organic matter content was positively correlated with soil infiltration [2]. Therefore, a stand with a high litter accumulation rate plays an important role in reducing soil bulk density and increasing porosity and infiltration of the soil [11].

Soil layer plays an important role in forest water storage; its water storage capacity was mainly determined by soil physical and chemical properties. In this study, soil water storage was higher in the mixed stand types than in the pure stand types and was higher in the broad-leaved stand types than in the coniferous stand types. Similar results were observed previously [14]. Cao et al. also found that the soil moisture under a broad-leaved deciduous forest was higher than that under an evergreen coniferous forest, most likely because of the high shading effect of the broad-leaved tree species, lowering soil evaporation [15].

Gray correlation analysis revealed that soil capillary and non-capillary water storage were the main factors determining the soil and water conservation ability of the stand because the growth and apoptosis of plant roots formed more pores in the soil, providing favorable conditions for water storage [11]. There is a close relationship among litter, soil, water, and stand, and water is the link among them. The stand returns the dead branches and leaves to the surface of the plantation; after the decomposition of litter, the nutrients are brought into the soil through rainwater leaching, which increases the soil nutrients and promotes the growth of trees [46]. While in the process of tree growth, the root growth and apoptosis make the soil loose and porous, which change the soil physical properties, increase the soil water holding capacity, promote the trees' growth, and then form more litters [11]. Therefore, water plays an important role in forest productivity, litter accumulation, soil physical and chemical properties, and soil water storage capacity. Comprehensive

evaluation of the soil and water conservation ability of the five stands showed that the mixed stand type was optimal, followed by broad-leaved stand types and, finally, coniferous stand types. These differences were mainly related to the accumulation, decomposition, and water holding capacity of stand litter and the degree of soil improvement by tree species. The results imply mixed stand types should preferentially be planted to enhance water and soil conservation in the construction of plantations in the northern and southern mountains of Lanzhou. In addition, planting shrubs and herbs under the plantation could increase soil porosity and achieve the maximum water holding capacity for the stand.

## 5. Conclusions

The results of this study indicate that Pa + Rp had the highest accumulation and maximum water holding capacity of litter, whereas the Pa had the lowest. The soil maximum water storage was highest in Pa + Rp and lowest in Po; and the maximum water storage of the mixed stand types was greater than that of the pure stand types. The soil and water conservation function of the five stand types increased in the order of Po < Pa < Rp < Po + Rp < Pa + Rp. The main factors affecting the water and soil conservation of a stand were identified as soil capillary water storage, non-capillary water storage, and bulk density.

**Author Contributions:** S.S., data curation and formal analysis; X.L., writing—review and editing. All authors have read and agreed to the published version of the manuscript.

**Funding:** This research was supported by the Discipline construction fund of Gansu Agricultural University (GAU-XKJS-2018-103; GAU-XKJS-2018-102), National Natural Science Fund (32060335), and Regular science and technology assistance projects for developing countries (KY202002011).

**Institutional Review Board Statement:** Not applicable.

**Informed Consent Statement:** Not applicable.

**Data Availability Statement:** Not applicable.

**Conflicts of Interest:** The authors declare they have no conflict of interest.

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
