# Peer review of "The Water Storage Function of Litters and Soil in Five Typical Plantations in the Northern and Southern Mountains of Lanzhou, Northwest China"

_sustainability, doi:10.3390/su14148231_

Round 1

Reviewer 2 Report

Peer review on manuscript «The Water Storage Function of Five Typical Plantations in the Northern and Southern Mountains of 2 Lanzhou, Northwest China»

The subject of the article fits within the scope of the journal. The study focuses on the study of water-regulating function of 5 Typical Plantations in the Northern and Southern Mountains of 2 Lanzhou (China).

The manuscript is well structured and the conclusions are clear. The illustrative material is sufficient and adequately reflects the results obtained by the authors. The manuscript is well structured, the conclusions are clear. The illustrative material is sufficient and adequately reflects the results obtained by the authors. In general, the article contains very interesting material of scientific value.

However, after reading the manuscript, I had some questions, answering which the article can be accepted for publication in the journal "Sustainability".

1. The shrub layer is well developed in mixed forests as part of the forest stand. What is its water holding capacity? Or is its participation incomparable with the role of woody plants? This was not mentioned in the results and discussion.

2.  In section 3.3, line 144, the authors refer to table 4. However, there is no such table in the manuscript. This is mistake?

3.     In figure 2, the graphs on the right are not readable, they do not show any differences by forest type. You probably need to change the scale on the y-axis.

4.     When describing the results obtained, the authors first give a range of variation from a smaller to a larger value. and then list the forest types in descending order. In my opinion, it is more logical then to bring the types in ascending order. This will make the material clearer and easier to understand.

 I think it will not be difficult for the authors to make corrections and give explanations to these comments. 

Reviewer 3 Report

The paper is well written, however, the paper is not consistent with the methodology and introduction. My comments are in the paper

Reviewer 4 Report

Refer to the attached file

Round 2

Reviewer 1 Report

Dear authors,

The manuscript had significantly increased its quality. Again, the head of figures and tables should be self explanatory, that means you do not have to mention the species with an abbreviation. The scientific names must be completed. In figure 1 you have to mention the longitude and latitude coordinates.  

Author Response

Please see the attachment,Thank you.

Reviewer 4 Report

Please refer the attached file.
